# Environment-Friendly and Efficient Gaseous Insulator as a Potential Alternative to $SF_6$

**Hafiz Shafqat Kharal [1],\* , Muhammad Kamran [1], Rahmat Ullah [2] ,**
**Muhammad Zaheer Saleem [2],\* and Muhammad Junaid Alvi [1]**

[1]   Department of Electrical Engineering, University of Engineering and Technology, Lahore 54890, Pakistan;
      mkamran@uet.edu.pk (M.K.); alvi_junaid@yahoo.com (M.J.A.)
[2]   Faculty of Electrical Engineering, GIK Institute of Engineering Sciences and Technology, Topi 23640,
      Pakistan; rahmat.ullah@giki.edu.pk
\*    Correspondence: engr.hsakharal@hotmail.com (H.S.K.); zaheersaleem786@gmail.com (M.Z.S.)

**Abstract:** Sulfur hexafluoride ($SF_6$) is commonly used in electrical insulation networks due to its superior dielectric properties. However, it possesses a high Global Warming Potential (GWP) of 22,800 times compared to $CO_2$ (at equal mass over a time span of 100 years) and a high atmospheric lifetime. This alarming metric prompted investigation for substitute gases with minor environmental influences. The overall objective of this research is to evaluate refrigerant R152a as a potential alternative for $SF_6$ in electrical insulation systems. R152a gas has a significantly reduced value of GWP (140) and is a cheap insulation medium as compared to $SF_6$. In this paper, dielectric breakdown testing of R152a and a mixture of $CO_2$ with different concentrations have been tested. The dielectric strength of R152a/$CO_2$ gas shows a saturated growth trend with increasing the gap difference, gas pressure and mixing ratio of R152a. Based on our experimental conditions, R152a/$CO_2$ gas reveals good dielectric properties, and insulation performance can reach up to 96% of $SF_6$. Finally, this work will bring a cost-effective and environment-friendly gaseous insulator for utility companies and power equipment manufacturers.

**Keywords:** R152a/$CO_2$; global warming potential; breakdown characteristics; environment-friendly; insulating material; power frequency

## 1. Introduction

Increasing demand for efficient electrical energy use has become a challenge, especially for developing countries with their increased energy usage and poor planning for supply and maintenance. This unplanned and unpredicted exponential rise in energy demand has increased the requirement for deployment of better power protection systems that may withstand undue and unwanted system failures. Therefore, protection equipment should be installed with the best and efficient insulation medium to overcome quick faulty circuit isolation. The most widely used gaseous insulator is sulfur hexafluoride ($SF_6$) with superior arc quenching and insulation abilities [1]. $SF_6$ is a widely used gas and a very prevalent choice in insulation medium for high voltage (HV) apparatus, gas-insulated power lines, compact circuit breaker. It is also non-flammable as compared to the insulating mineral oils particularly for the indoor environments. However, possessing the aforesaid features and properties $SF_6$ has been regrettably found to be a damaging greenhouse gas stated by the Kyoto Protocol that it is 22,800 times more harmful than $CO_2$ for the same masses of both the gases over a time span of 100 years [2,3]. Therefore, $SF_6$ is included in the prohibited list for ecological safety. $SF_6$ decay products, as well as the moisture inside the apparatus, can cause damages to materials like alumina. Therefore, molecular filters were used for harmless exclusion and absorption purposes [4]. Earlier investigations

were carried out to expand $CF_3I/N_2$ [5–9], $C_5F_{10}O$ [10,11], $C_6F_{12}O$ [12], $C_3F_7CN$ [13] and $C_4F_7N$ [14] as an alternate of $SF_6$, but these candidates have some disadvantages described in Table 1. Gases cannot be maintained in a closed vessel, according to IEC 62271-1 standard there will be a 0.1% leakage from a closed vessel [15]. A worldwide 25-year accumulative data is calculated based on a 0.1% leakage ratio as shown in Table 2 [16,17].

**Table 1.** Association of $SF_6$ replacement.

| Gases | Problems and Drawbacks |
|---|---|
| Carbon dioxide, Nitrogen and Dry air | Momentous expansion in pressure. Momentous expansion in size of equipment. Low breakdown voltage [18]. |
| Trifluoro iodomethane mixtures (CF3I/$CO_2$ or $N_2$) | Boiling point large than that of $CF_3I$ (−22.5 °C) at 0.1 MPa. Classified as a perilous, mutagenic, and venomous for facsimile (Type-3) [19]. |
| Mixtures of per-fluorinated ketones ($C_5F_{10}O$, $C_6F_{12}O$/Technical air or $CO_2$) | Superior smallest operating temperature than $SF_6$ [20]. Far above the ground boiling temperature (24 °C) at (0.1 MPa) because of higher molecular mass. |
| HFO 1234ze | Carbon grime dump on electrodes owing to high spark voltage. Superior operating temperature than $SF_6$ while unpolluted (constrained at −15 °C). |
| $C_4F_7N/CO_2$ | Having high boiling point (−4.7 °C at 0.1 MPa) [14] |

**Table 2.** Worldwide usage and leakage of $SF_6$ from all GIS.

| Worldwide $SF_6$ Insulated GIS | Expected Mass of Each Component | Total $SF_6$ Used in All GIS | Annually Leakage of $SF_6$ from all GIS | Cumulative 25 Years Leakage of $SF_6$ from All GIS |
|---|---|---|---|---|
| 20,000 | 500 kg | 10,000,000 kg | 9881 Kg | 392,944 Kg |

Such critical matters have pointed the emphasis to improve the fidelity and adeptness of power transmission and distribution systems around the world while ensuring that the upcoming modern innovations and latest technologies are not hazardous to the environment.

It is very important to reflect on the environment when designing electric power systems. Such investigation helps to formulate the development of an insulating medium that is environmentally and economically attractive. R152a is a new type of insulating and environmental-friendly gas that has been acknowledged by scholars from all over the world in the past three years. Moreover, a better understanding of the characteristics of R152a and their breakdown mechanisms is developed. The comparative analysis of different $SF_6$ alternatives has been shown in Table 3. Here in Table 3 dielectric strength of all alternatives was normalized to $SF_6$ at 1 bar.

R152a is recognized as a chlorofluorocarbon (CFC), which is commonly used in refrigeration appliances and in aerosol sprays with properties in compliance with the Montreal Protocol [21]. R152a possesses some pertinent qualities making it an effective gas to be employed in the field; for example, it is harmless and non-explosive. All these features make it a suitable candidate for domestic and industrial usage as an electric insulator. R152a gas has a significantly reduced value of GWP (140) and is a cheap insulation medium compared to $SF_6$. Moreover, R152a has zero ozone depletion potential. As the atmospheric lifetime of R152a is 1.4 years so its decomposition products have 98% low environmental impact as compared to $SF_6$. Therefore, using the proposed gas mixture can effectively reduce the greenhouse effect. Table 4 shows a contrast between the physical and chemical properties of R152a and $SF_6$.

**Table 3.** Different alternative of $SF_6$ gas. The global warming potential represents the values over a time span of 100 years for equal masses of these gases.

| Reference | Gas | Dielectric Strength (DS) | Global Warming Potential | Atmosphere Lifetime | Boiling Point | Cost/kg |
|---|---|---|---|---|---|---|
| [11] | $SF_6$ | 1 | 22,800 | 3200 | −63 °C | 25–30 \$ |
| [22] | $N_2$ | 0.40 | 0 | | −195.8 °C | 0.25 times of $SF_6$ |
| [23] | $CO_2$ | 0.37 | 1 | | −78.5 °C | 0.35 times of $SF_6$ |
| [23] | $C_2F_6$ | 0.80 | 12,200 | 10,000 | −78.1 °C | 2.5 times of $SF_6$ |
| [23] | $C_3F_8$ | 0.90 | 8830 | 2600 | −36.7 °C | 2 times of $SF_6$ |
| [23] | $CF_3I$ | 1.21 | 5 | 0.05 | −22.5 °C | 10 times of $SF_6$ |
| [23] | $C_4F_{10}$ | 1.2–1.3 | 8700 | 3200 | −5.99 °C | 9 times of $SF_6$ |

**Table 4.** Contrast between physical and chemical properties of $SF_6$ versus R152a [20]. The global warming potential represents the values over a time span of 100 years for equal masses of these gases [24–26].

| Properties | $SF_6$ | R152a |
|---|---|---|
| GWP | 22,800 | 140 |
| Density | 6.17 kg/m$^3$ | 2.7 kg/cm$^3$ |
| Molecular mass | 146.06 g/mol | 66.1 g/mol |
| Atmospheric life | 3200 | 1.5 |
| Boiling point | −64 °C | −25 °C |
| Appearance | Colorless | Colorless |
| Electronegativity | 2.5 | 2.42 |
| Price/kg | \$28 to \$30 | \$12 |

Presently, achievements have been made in the dielectric properties of the R152a/$CO_2$ gas mixture. An experimental study on the liquefaction temperature reduction and power frequency breakdown characteristics of R152a/$CO_2$ (different mixtures under different pressure and varying gap distance) was performed. Further, the experimental setup, testing methods, procedures and GWP of R152a/$CO_2$ mixture are elaborated in this paper.

## 2. Laboratory Test Setup and Assembly of Test Electrodes

This section describes the experimental test setup in the high voltage laboratory, techniques applied and assembly of test electrodes. Figure 1 shows the schematic for performing the breakdown voltage testing according to the IEC60270 standard [27]. The test setup consists of a control desk (HV-9103) comprising a peak voltmeter (HV 9150) and built-in variable voltage supply. The power supply output is 0 to 230 volts and the voltmeter peak range is 100 to 1000 (kV). The control desk consists of measuring instruments, namely Impulse, Peak, Trigger devices and DC Voltmeters. For DC breakdown testing, through rectification 140 kV DC was generated.

As the breakdown occurs, the voltmeter measures the breakdown voltage across the measuring capacitor. Prior to starting the experimental breakdown voltage tests, AC Voltage of identified value was applied to a voltmeter and measuring devices for calibration purposes to minimize errors and improve precision. All experimental results were obtained at room temperature (20–25 °C). The pressure vessel's temperature rise has been avoided by the interval of 5 min between two consecutive breakdowns as recommended in [14].

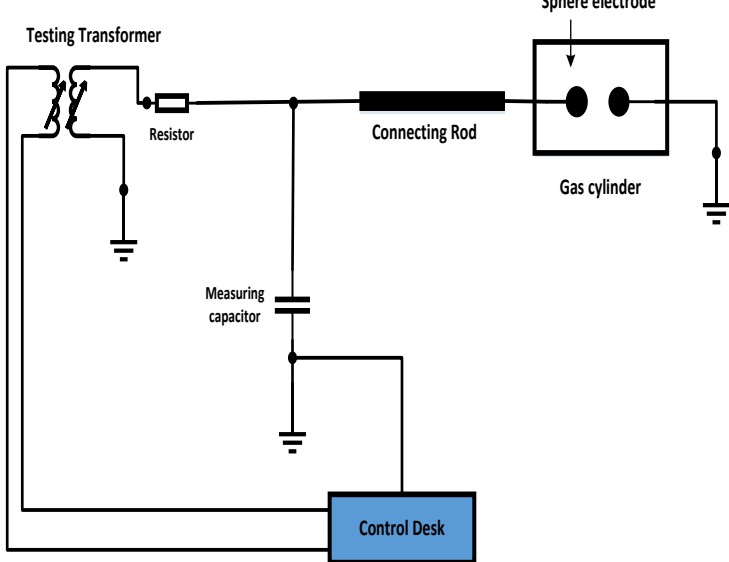

**Figure 1.** Schematic to examine R152a/$CO_2$ breakdown voltage by sphere–sphere electrodes.

Testing vessel for vacuum and gas is made of steel and equipped with a pressure gauge to measure pressure up to 6 bars. The manufacturing material of the electrode was aluminum enclosed with a nickel coating. The diameter of the electrodes is basically 50 mm. Electrode diameter was selected to be 50 mm because the gap length should be equal or less than the radius of electrode to maintain uniform electric field. In our experimental work the gap length is varied from 0–16 mm, therefore 50 mm is best appropriate diameter of electrodes. Figure 2a,b show experimental setup and testing vessel (HV-9134) respectively. The vessel contains a cylinder made of Plexi-glass that is sandwiched with flanges top and bottom which are linked with high voltage (HV) and ground potential correspondingly. The bottom cover is furnished with essential apparatus, such as inlet, outlet valve measuring gauge for vacuum and pressure. The specifications of the test vessel provided by the manufacturers are briefly described in Table 5.

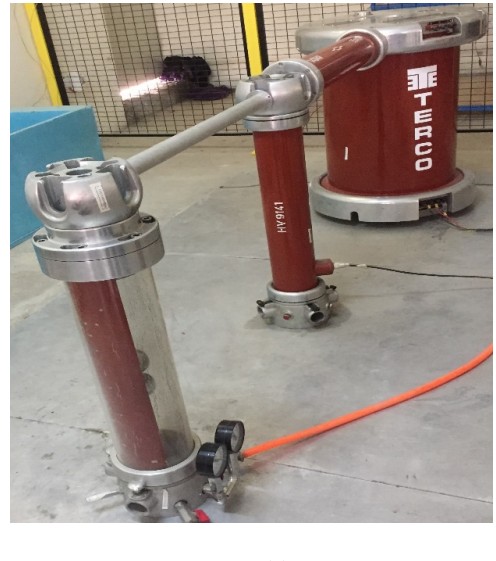

(**a**)

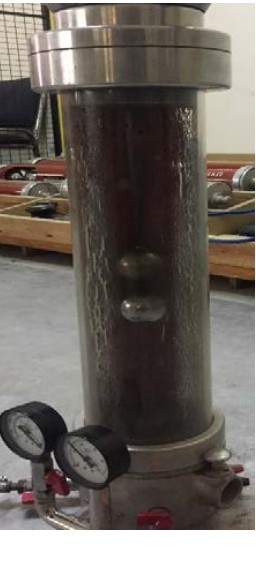

(**b**)

**Figure 2.** Test equipment used: (**a**) experimental setup; (**b**) testing vessel (HV-9134).

**Table 5.** Test setup specification.

| Specifications | Standards |
|---|---|
| Voltage (AC) | 100 kilovolts |
| Pressure (p) | 0 to 6 bars |
| Diameter of sphere electrodes | 50 mm |
| Vertical height | 800 mm |

## 3. Power Frequency Breakdown Voltage Experiments

### 3.1. Experimental Procedure

Prior to start testing, both electrodes were cleaned with alcohol dumped silk textile cloth to eliminate moisture and impurities to minimize errors and maximize accuracy in all observations. Tests were carried in dried and moisture-free zone at room temperature. An increase in temperature raises the probability of errors in experimental results. To overcome this problem, time span for each test was restricted to 15–20 min. As R152a and CO2 both are inert and in gaseous form, the time equal to 30–45 min is enough to mix properly for both gases [20].

### 3.2. Gas Mixture Procedure

Considering the proposed alternate gas mixture liquefaction temperature experimental constraints and different mixture ratios for power frequency breakdown voltages were mentioned in Tables 6 and 7.

**Table 6.** Experimental constraints.

| Configuration of Electrodes | Sphere–Sphere |
|---|---|
| Length of spark gap | 6 mm–18 mm |
| AC voltage | 0–100 kV (AC) |
| DC voltage | 0–140 kV (DC) |
| Material of electrode | Aluminum Ni plated steel |

**Table 7.** Different mixture ratio of R152a and $CO_2$.

| Measurement No. | R152a Ratio (%) | $CO_2$ Ratio (%) |
|---|---|---|
| 1 | 90 | 10 |
| 2 | 80 | 20 |
| 3 | 70 | 30 |
| 4 | 60 | 40 |
| 5 | 50 | 50 |

### 3.3. Calculation of Accurate Gas Mixture Pressure

In order to fill up the accurate amount of R152a and $CO_2$ to achieve the accurate ratio of the gas mixture by P/P, the calculation of the amount of gas required is essential. A notable thing here is the P/P ratios of the mixtures of gases because using the W/W ratio would render the calculations incorrect as the molar mass of molecules can change the pressure of the gas mixture heavily. The total amount of R152a and $CO_2$ needed is calculated using the ideal gas law as shown in Equation (1) [25].

$$V = \frac{mRT}{MW \times P} \tag{1}$$

where:

m = Mass of gas (grams), T = Temperature (Kelvin), P = Pressure (bar);
MW = Molecular weight of gas (g/mol), R = Ideal gas constant, V = Volume (liters).

For example, in R152a operation, the filling temperature was 20 °C. The ideal gas constant (R) is 0.0821, the MW of R152a is 66.01 g/mol and MW of $CO_2$ = 44.01 g/mol. Therefore, when the chamber is filled with mixture the volume of each gas can be calculated as follows [28]:

$$V = \frac{mRT}{MW \times P} = 1300 \times 0.0821 \times 293/146 \times 1.4 = 153 \text{ L}$$

In order to fill 80% R152a required amount of this gas

$$m = \frac{MW \times P}{RT} = (66 \times 0.98 \times 153)/(0.0821 \times 293) = 412 \text{ g}$$

Similarly, for 20% amount of $CO_2$

$$m = \frac{MW \times P}{RT} = (44 \times 0.42 \times 153)/(0.0821 \times 293) = 117 \text{ g}$$

Therefore, the amount required to fill 80%/20% mixture ratio of R152a/$CO_2$ is 412:117 g.

### 3.4. Mixture Ratio Analysis

Experiments were performed to locate Alternating Current (AC) power frequency breakdown characteristics on 8 mm electrode distance under these environments (a) pure R152a, (b) pure $CO_2$, (c) $CO_2$ (50%) with addition of R152a (50%), (d) $CO_2$ (40%) with addition of R152a (60%), (e) $CO_2$ (30%) with addition of R152a (70%), (f) $CO_2$ (20%) with addition of R152a (80%) and (g) R152a (10%) with addition of R152a (90%). Figure 3 shows the breakdown strength of R152a and CO2 among their mixture at different R152a/$CO_2$ ratios.

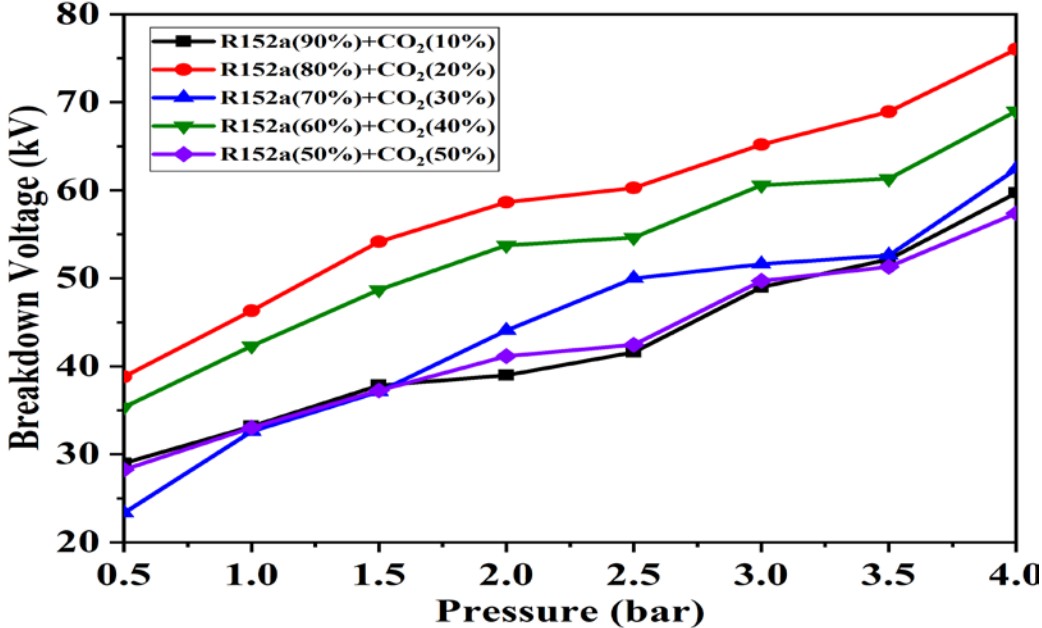

**Figure 3.** AC power frequency breakdown voltage of R152a/$CO_2$ gas at varying mixture ratio and 8 mm electrode gap distance.

R152a is an electronegative gas, and all negative ions are created by gaining electrons from neutral molecules R152a, which as a result become positive ions after losing electrons. Gaining and losing of the electrons could occur depending on the field applied and attachment and detachment capability of the insulating medium. Losing electrons or detachment coefficients is symbolized by η as shown

in Equation (2). When a single electron travels per unit length, several electrons produced in that specified path are defined by Townsend first ionization coefficient, $\alpha$.

$$dN = N\,(\alpha - \eta)\,dx \tag{2}$$

where N refers to initial electron quantity, dN denotes the no of electron traveled a distance dx.

### 3.5. Dielectric Strength Analysis

The breakdown strength of R152a and $CO_2$ different mixtures was measured in uniform field under AC voltage. Figure 4 displays the dielectric strength characteristics which can be attained by ratios of 80% and 20% respectively on 4 bar which gives the highest breakdown strength of 96% of $SF_6$ gas.

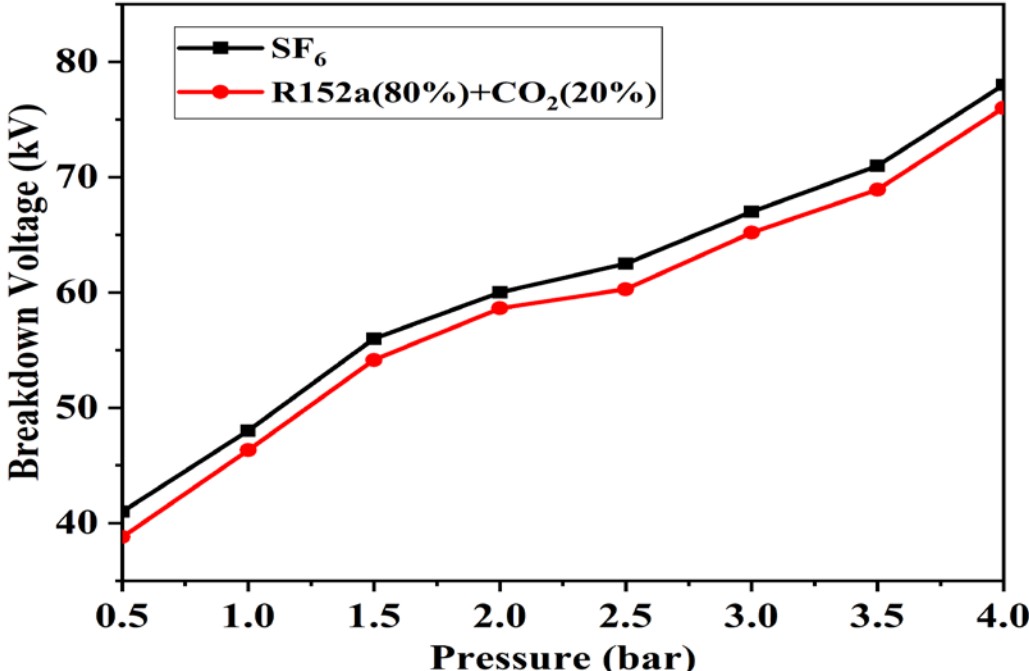

**Figure 4.** Breakdown characteristic comparison of R152a/$CO_2$ gas at 80%/20% mixture ratio and $SF_6$ at 8 mm electrode gap distance.

### 3.6. Gap Difference Analysis

The breakdown voltage of R152a/$CO_2$ gas varied with the electrode gap distance (4–16 mm). This gap between both electrodes has dominant effects on the gas dielectric strength as shown in Figure 5. In Equation (3) there is an almost linear relationship between the electrode gap and breakdown voltage can be seen [29].

$$E = f \times (V/D) \tag{3}$$

where constant f is demonstrating non-uniformity, V is applied voltage and D is the distance between two electrodes. R152a/$CO_2$ (80%/20%) revealed a similar growth trend as $SF_6$ by changing the gap length as shown in Figure 5. After 12 mm, no significant improvement in breakdown voltage was found for other mixtures of R152a/$CO_2$.

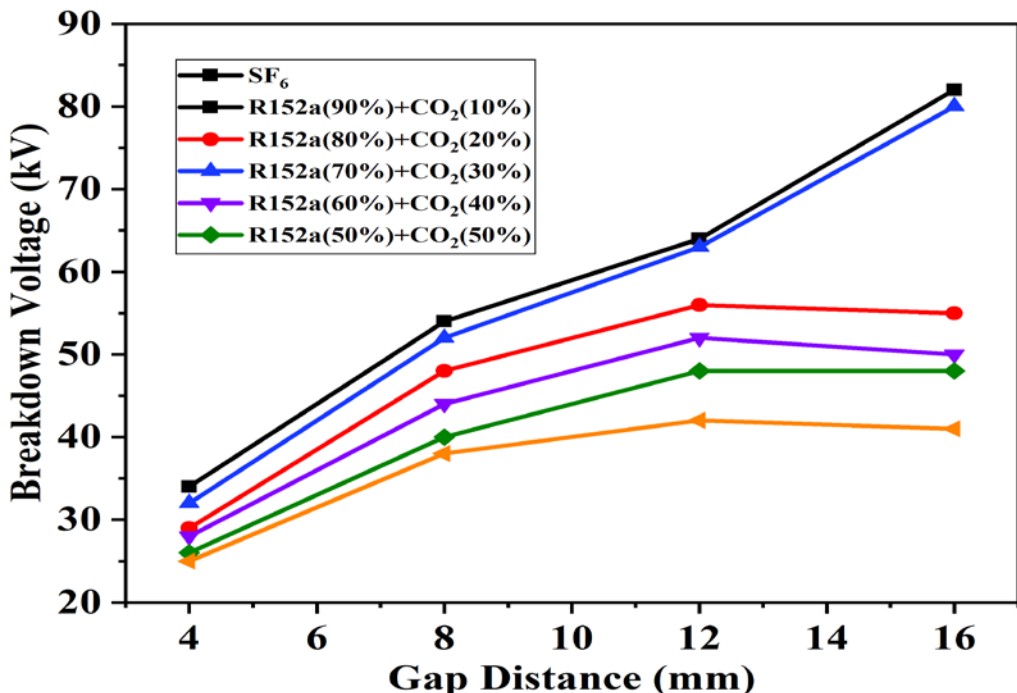

**Figure 5.** Breakdown voltages of R152a/CO$_2$ gas varying the gap distance (4–16 mm) at different mixture ratio.

*3.7. Statistical Analysis of R152a*

Table 8 demonstrates the statistical analysis of R152a along with CO$_2$. These are premeditatedly designed for variable magnitudes of two mixed gases. The experimental values like standard deviation (SD), coefficient of variation and mean deviation are also depicted in mentioned Table 8 that demonstrates the inconsistency of results achieved through the experiment. The SD of R152a and CO$_2$ (70%/30%) shows a rapid variation in value. Correspondingly, coefficient of variation is observed at the lowest value at (90%/10%) mixing ratio and the mean deviation was found lowest value at (60%/40%) in Table 8.

**Table 8.** Statistical analysis of R152a.

| Base Gas R152a Mixed Gas CO$_2$ | | | | | |
|---|---|---|---|---|---|
| RBG [1] | 50% | 60% | 70% | 80% | 90% |
| SD | 11.2 | 12.3 | 10.21 | 14.2 | 12.9 |
| M | 46.1 | 51.6 | 59.7 | 57.3 | 55.3 |
| cv | 0.23 | 6 | 0.27 | 0.28 | 0.19 |
| Max kV | 60 | 69.8 | 73 | 76.1 | 71.6 |
| Min kV | 26 | 4 | 49.8 | 43 | 35.6 |

[1] RBG: Ratio of base gas; SD: standard deviation; M: mean; cv: coefficient of variation.

## 4. HVDC Analysis

Figure 6 shows the breakdown voltage of R152a/CO$_2$ under HVDC. HVDC in the range of 0–140 kV was generated by the Greincher voltage doubler circuit. The best breakdown strength in case of HVDC was achieved for R152a/CO$_2$ (80%/20%).

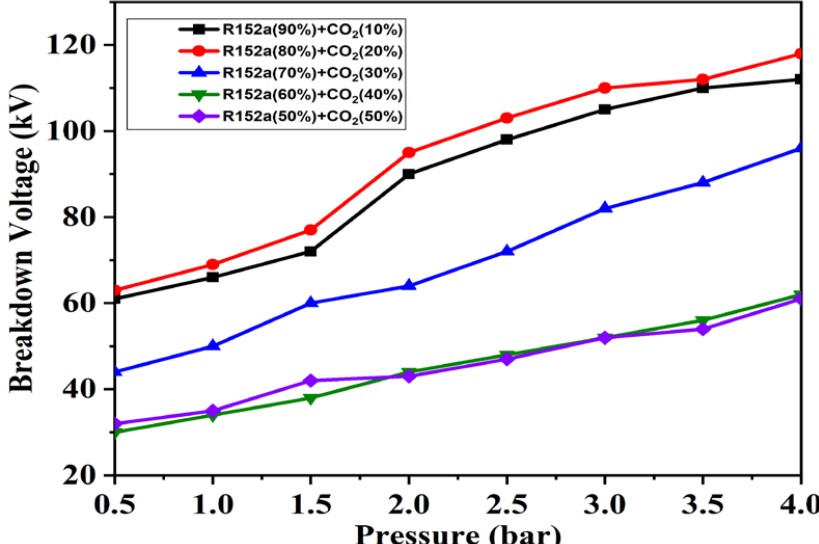

**Figure 6.** DC breakdown voltage characteristics of R152a/CO$_2$ with different mixture ratios and 8 mm gap distance.

## 5. Global Warming Potential (GWP) Analysis

This novel alternative R152a/CO$_2$ gas mixture has been particularly developed to significantly reduce GWP as compared to SF$_6$. According to environmental protection view, the GWP is calculated as a weighted average of this proposed gas mixture, and from the sum of the weight fractions of each substance and multiplied with their individual GWP as given in Equation (4) where k shows the base gas mixing ratio, 140 is the GWP of R52a and 44 and 56 is the molar mass of R152a and CO$_2$ respectively [14].

$$GWP = \frac{k \times 56 \times 140 + (1-k) \times 44 \times 1}{k \times 56 + (1-k) \times 44} \tag{4}$$

The relationship between the GWP value and mixing ratio is shown in Figure 7. It is found that R152a/CO$_2$ mixture contents with ratio 80%/20% at −14.16 °C has a total GWP of 117.17 instead of 22,800 of SF$_6$ over a 100-year time span, effectively reducing the greenhouse effects by 98%.

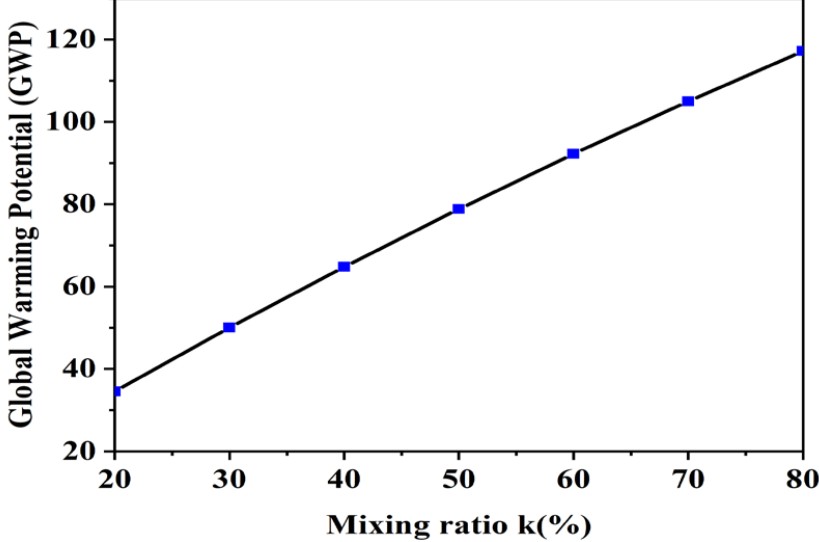

**Figure 7.** Global Warming GWP analysis of R152a/CO$_2$ gas mixture.

## 6. R152a/CO$_2$ Liquefaction Temperature Analysis

In the practical engineering application of new alternate gas, the most important parameter is liquefaction temperature limitation. R152a liquefaction temperature is −25 °C, while that of SF$_6$ is −64 °C. The relationship of vapor pressure with condensation temperature is shown in Figure 8 for R152a and SF$_6$ revealed lower value of condensation temperature for R152a. Thus, it becomes necessary to mix R152a with air, CO$_2$, N$_2$ or buffer gases to meet the requirement of low liquefaction temperature. CO$_2$ has been preferred over other buffer gases like nitrogen and air due to its superior arc quenching ability to produce the appropriate mixture for circuit breaker and disconnect switch applications [30]. In this research, R152a was used along with the mixtures of CO$_2$ resulting in reduced depletion of the ozone layer and acceptable liquefaction temperature.

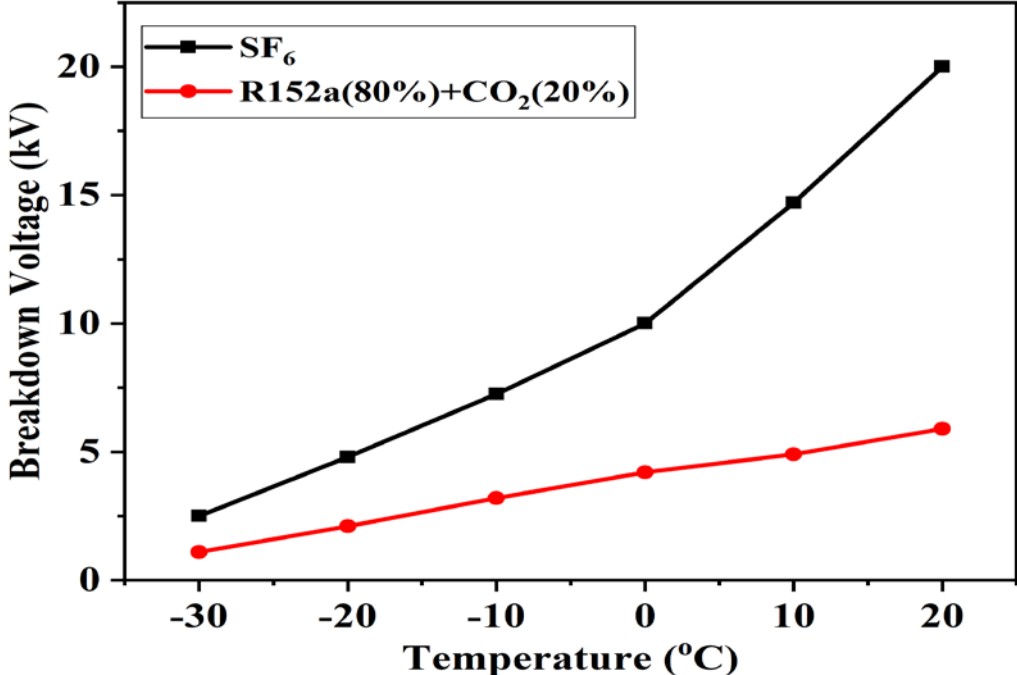

**Figure 8.** Saturated vapor pressure R152a and SF$_6$.

The formula for calculating the liquefaction temperature or saturated vapor pressure has been given in Equation (5) [31]. The boiling point (b.p) of R152a is greater than SF$_6$ (−63 °C). Due to this reason, buffer gas CO$_2$ is added to reduce the disadvantages of high boiling point because CO$_2$ possess very low b.p. Increasing the CO$_2$ content in the mixture of R152a/CO$_2$ reduces overall b.p. Figure 9 shows mixed gas liquefaction temperature. It was proposed that by increasing the ratio of additive gas, the overall liquefaction temperature was reduced. The characteristic curve was achieved by assuming ideal additive gases.

$$P = \exp[A\left(\frac{1-\frac{T_b}{T}}{R}\right)] \tag{5}$$

where P represents the gas boiling point pressure; T is the liquefaction temperature of R152a in mixture with CO$_2$; Tb is the liquefaction temperature (K) at atmospheric pressure; R = 2 cal/deg.mol is the gas constant and A = 21 cal/deg.mol is constant.

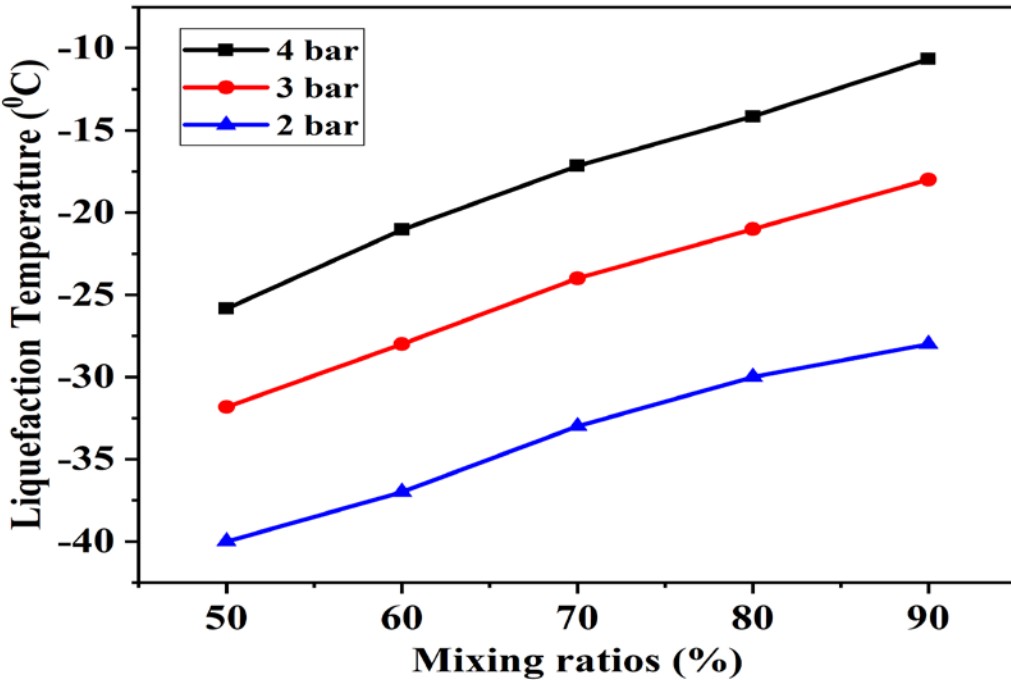

**Figure 9.** Liquefaction temperature of R152a at different pressure and mixture ratio.

## 7. Conclusions

This paper emphasizes on dielectric properties of R152a/$CO_2$ in different mixture ratios as a potential $SF_6$ substitute for engineering applications and provides guidance for future research. This gas mixture was experimentally investigated and compared with $SF_6$ under power frequency breakdown characteristics, effect of different mixture ratio, pressure and gap distance. GWP is also examined which provides promising results. Considering the liquefaction condition, the following conclusions can be summarized through extensive laboratory testing:

(1) The insulation strength of gas mixture R152a/$CO_2$ (80%/20%) can reach more than 96% of $SF_6$. This enabled us to present the optimal ratio achieved by replacing $SF_6$ as the insulation performance of R152a/$CO_2$ (80%/20%) is close to $SF_6$.

(2) The AC breakdown voltages of R152a/$CO_2$ increase linearly by increasing the gap length. The proposed gas mixture demonstrates good dielectric properties by increasing the content of R152a with comparatively low-temperature applications.

(3) Furthermore, these formulated mixtures are cost-effective and reduce the amount of GWP 98% as compared to pure $SF_6$.

R152a/$CO_2$ is concluded to be a potential proposed composite replacement gas for electrical applications. Therefore, this environment-friendly alternative should be put into practice to save energy and improve the performance of the system.

**Author Contributions:** Conceptualization, H.S.K. and M.K.; methodology, M.J.A. writing—original draft preparation, H.S.K.; writing—review and editing, M.K., M.Z.S. and R.U.

**Funding:** This research was funded by Higher Education Commission, Pakistan, grant number 50018514 and the APC was also funded by HEC, Pakistan.

**Acknowledgments:** The authors thank the higher education commission of Pakistan for financial support to this work's commencement. A statement of grateful acknowledgment goes to our most respected Rasheed (late) from COMSATS Pakistan for his guidance, project work, and laboratory support.

**Conflicts of Interest:** The authors declare no conflict of interest.

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
