# Peer review of "Environment-Friendly and Efficient Gaseous Insulator as a Potential Alternative to SF6"

_processes, doi:10.3390/pr7100740_

Round 1

Reviewer 1 Report

This paper proposes a combination of R152a and carbon dioxide, as an alternative of Sulphur hexafluoride (SF6) acting as the insulation medium of high voltage equipment. The breakdown voltage of the R152a was tested under different gas mixture conditions, pressure and gap difference. The results of dielectric tests showed a good performance of the proposed insulation medium. The content of the paper can be further improved by following comments:

The quality of the figures need to be improved as most figures are blur.

Refer to the title of the paper, the environment impact of the proposed gas mixture as an insulation medium needs to be discussed.

The potential applicability of the proposed gas mixture as an insulation medium in real practice needs to be elaborated.

There are some grammatical errors (e.g., noun-verb agreement, improper capitalization) that need correcting. Please spend some time editing this paper before resubmission.

Reviewer 2 Report

The authors presented an alternative to SF6 gas in the form of a gas mixture of R152a/CO2. According to the author, such a mixture has good properties which can be used in high voltage equipment. The most important properties are high breakdown voltage and much lower Global Warming Potential in comparison to SF6. However, the authors of the article omitted a very important issue related to the flammability of R152a. In this aspect, the issue of the influence of temperature and pressure on the liquefaction of this gas is very important. The authors of the article should refer primarily to these issues.
In my opinion, the research results presented by the authors are valuable. However the article has some shortcomings and needs to be improved before being accepted for publication in Processes.

Dear authors, please refer to the following suggestions and comments:

Major comments

Lines 90-91: Why did you choose for the research the electrodes of the diameter of 20 mm?

Line 73: Did you control temperature during laboratory tests?

Line 101: Please describe the parts of measuring setup in Figure 2c

Line 109: Do you think that time equal to 30-45 minutes was enough to mix R152a and CO2 properly. You selected this time on the basis of literature where CO2 and fluoronitriles were mixed.

Line 109: There is a difference between the density of R152a and CO2 are you sure that mixture consists of these two gases is long time stable. What is the influence of the gas pressure on the stability of this gas mixture?

Line 180: Could you explain why for the gas ratio 80%R152a/20%CO2 we observe breakdown voltage increase also for the gap distance above 12 mm? – in case of other gas ratios breakdown voltage didn’t change significantly

Do you think that flammability of R152a is a restriction on the use of this gas in high voltage equipment?

Line 150: you should analyse in chapter 7 how the presence of CO2 in the mixture with R152a may improved liquefaction temperature.

Could you compare in your article the properties of your R152a and CO2 gas mixture and described in literature mixture of Fluoronitrile/CO2

Native speaker proofreading is necessary.

Minor comments

Line 6: „laboratory” instead of „laboratory” use a capital letter

Line 51: incorrect font

Line 76: Incorrect reference, there is no information about IEC60279 in [22]

Lines 82 and 84: In my opinion references [23] and [24] are not necessary or are cited in an improper way.

Line 115: „in Table” instead of „In Table”

Line 133: 44.01 g/mol” instead of 44.01 g/”

Reviewer 3 Report

The manuscript by Kharal and coworkers is based upon the Global Warming Potential (GWP) of SF6 claimed to be 23800 times greater than that of CO2. The reference for this metric given in the manuscript is the “Kyoto protocol”. The authors seem to ignore than this value is by no means a constant but is strongly dependent upon the actual or potential concentration of SF6 in the atmosphere. The GWP value is high nowadays because the concentration of SF6 is extremely low, fortunately. If the concentration of SF6 in the atmosphere would become similar to that of CO2, 0.04 %, the GWP of SF6 would be of the same order of magnitude as that of CO2, viz. little above 1, because the difference of electronegativity of the S-F chemical bounding, although higher, is not very different of that of the C-O bond. The difference of electronegativity controls the infrared absorption of the molecule, hence the greenhouse gas effect. The reason for the strong concentration dependence of GWP is that the derivative of a logarithmic function, relevant for the impact of a greenhouse gas, is a hyperbolic function. As a result, when the concentration in the atmosphere is extremely low, the dependence on concentration is large.

As a result, the calculation of the GWP of a gas mixture by considering the GWP of SF6 as a constant in paragraph 5 is questionable, and even appears of dubious validity.

The referee recommends that the authors consider the points above and resubmit a version of their manuscript revised along the lines suggested.    

Round 2

Reviewer 1 Report

The reviewer has no further comments.

Reviewer 2 Report

Dear authors

Thank you for answering all my questions and improving the article.

The quality of Figures 1 and 2 should be improved.

I don't have any more questions.